# Ecofriendly Removal of Aluminum and Cadmium Sulfate Pollution by Adsorption on Hexanoyl-Modified Chitosan

**Berthold Reis** [1,†], **Konstantin B. L. Borchert** [1,†], **Martha Kafetzi** [2], **Martin Müller** [1,3], **Karina Haro Carrasco** [1], **Niklas Gerlach** [1], **Christine Steinbach** [1], **Simona Schwarz** [1,*], **Regine Boldt** [1], **Stergios Pispas** [2] and **Dana Schwarz** [1,*]

1 Leibniz-Institut für Polymerforschung Dresden e. V., Hohe Straße 6, 01069 Dresden, Germany
2 Theoretical and Physical Chemistry Institute, National Hellenic Research Foundation, 48 Vassileos Constantinou Ave., 11635 Athens, Greece
3 Department Chemistry and Food Chemistry, Technische Universität Dresden, 01062 Dresden, Germany
\* Correspondence: simsch@ipfdd.de (S.S.); schwarz-dana@ipfdd.de (D.S.); Tel.: +49-351-4658-333 (S.S.); +49-351-4658-542 (D.S.)
† These authors contributed equally to the work.

**Abstract:** The purity and safety of water as a finite resource is highly important in order to meet current and future human needs. To address this issue, the usage of environmentally friendly and biodegradable adsorbers and flocculants is essential. Chitosan, as a biopolymer, features tremendous properties as an adsorber and flocculant for water treatment. For the application of chitosan as an adsorber under acidic aqueous conditions, such as acid mine drainage, chitosan has been modified with hydrophobic hexanoyl chloride (H-chitosan) to reduce the solubility at a lower pH. In order to investigate the influence of the substitution of the hexanoyl chloride on the adsorption properties of chitosan, two chitosans of different molecular weights and of three different functionalization degrees were analyzed for the adsorption of $CdSO_{4(aq)}$ and $Al_2(SO_4)_{3(aq)}$. Among biobased adsorbents, H-chitosan derived from the shorter Chitosan exhibited extraordinarily high maximum adsorption capacities of 1.74 mmol/g and 2.06 mmol/g for $Cd^{2+}$ and sulfate, and 1.76 mmol/g and 2.60 mmol/g for $Al^{3+}$ and sulfate, respectively.

**Keywords:** modified chitosan; heavy metal ion adsorption; cadmium and aluminum removal; pH stable chitosan; wastewater treatment

## 1. Introduction

The contamination of the aquatic environment with metal ions, such as cadmium and aluminum, is of great concern because of their severe toxicity and lack of biodegradation in nature [1]. The increasing demand for the decontamination of wastewater and surface water has stressed the development and testing of natural adsorbers and flocculants [2–4], as synthetic polymers themselves risk toxicity due to, e.g., unreacted monomers from the synthesis, and are often non-biodegradable. In particular, ecofriendly and non-toxic bioadsorbents based on pectin [5,6], starch [7–9], cellulose [6], food waste [10], and chitosan [2,11] have been investigated for use in wastewater treatment, as they present a potential application for the remediation of natural water bodies. Among these materials, chitosan exhibits outstanding adsorption capacities for most toxic metal ions, which makes it ideal for even highly polluted waters [12–15]. However, the removal of heavy metal ions from acidic aqueous media such as acid mine drainage with chitosan is extremely challenging due to the dissolution of chitosan in acidic media [14]. In order to translate the advantageous adsorption properties of chitosan in neutral waters to these acidic conditions, the amino side groups of the chitosan can be partially substituted with hexanoyl chloride. The hydrophobic side chains lead to a reduction of chitosan solubility by decreasing the number of protonable amino functionalities and the formation of hydrophobic domains,

thereby allowing the application of chitosan as a biobased polymer for the adsorption in acidic media. The synthesis of hexanoyl-modified chitosan has been thoroughly investigated, and it is known to be both biodegradable and biocompatible [16,17]. Nevertheless, it has never been tested for ion adsorption from acidic waste waters.

However, the amino functionalities are the dominating functional groups for the outstanding adsorption properties. In order to investigate the influence of the partial substitution of the amino groups with hexanoyl chloride on the adsorption properties of chitosan, different ratios of hexanoyl chloride were added to the reaction mixture, leading to diverse substitution degrees of the hexanoyl chitosan (H-chitosan). The adsorption properties of the H-chitosan samples were investigated with $CdSO_{4(aq)}$ and $Al_2(SO_4)_{3(aq)}$. Aluminum is present in our everyday life as, e.g., food container material, and is used in high quantities as a flocculant in water treatment. Especially for the application as a flocculant in water treatment, the complete removal from $Al^{3+}$ in water is aspired towards due to its cytotoxicity and neurotoxicity even at low concentrations. These are caused by the increased oxidative stress and altered enzyme activity for humans [18,19]. Thus, the World Health Organization proposed a guideline value as low as 0.2 mg/L for $Al^{3+}$ in drinking water. Besides its high acidity, $Al^{3+}$ poses severe risks in aquatic ecosystems to, e.g., fish (respiratory and ionoregulatory effects, and altered enzyme activity) and algae (altering the community structure and causing a potential reduction in phosphorous uptake) [20–22]. $Cd^{2+}$ easily accumulates in all of the organisms exposed to it, such as vertebrates, invertebrates and insects [18,23]. Thus, a thorough removal of $Al^{3+}$ and $Cd^{2+}$ from water is decisive.

Here, we synthesized six different hexanoyl chitosan samples (H-chitosan) [24] by varying both the molecular weight of the chitosan and the used amount of the functionalization agent hexanoyl chloride. In order to investigate the degree of substitution (DS%) and the chemical composition, all of the samples were analyzed by $^1$H-NMR and FTIR. Furthermore, nitrogen sorption measurements were conducted to investigate the dry-sate properties. The solvated state was examined using DLS and zeta potential measurements. Furthermore, we tested the adsorption of the H-chitosan toward toxic metal ions removal ($Cd^{2+}$, $Al^{3+}$) from the respective sulfate salt solutions. The obtained adsorption isotherms were fitted by different adsorption models (i.e., Langmuir, Sips and the Dubinin–Radushkevich model). Furthermore, we subsequently analyzed the adsorbents by SEM-EDX to gain insights into the adsorption mechanism.

## 2. Materials and Methods

### 2.1. Materials

#### 2.1.1. Chemicals used for the Modification of Chitosan

Ch90/60/A1 and Ch90/200/A1 originate from crab shells, and were purchased from Biolog Heppe® GmbH (Landsberg, Germany). Both polymers were used without further purification. The indices indicate a deacetylation degree of 90%, a viscosity of 60 and 200 mPas, and a 1% ash content, respectively. The molar masses correlate with the viscosity, and are 90–150 kDa for Ch90/60/A1 (given by the provider) and 620 kDa for Ch90/200/A1 (AF4 measurement Đ = 6.25, recently published [25]).

The hexanoyl chloride and the sodium hydroxide were purchased from Sigma Aldrich (Merck KGaA, Darmstadt, Germany), concentrated acetic acid was purchased from Merck KGaA (Darmstadt, Germany), and acetone was purchased from VWR (99%, Darmstadt, Germany). All of the chemicals were used without any further purification.

#### 2.1.2. Heavy Metal Salts

The heavy metal salts $Al_2(SO_4)_3$ and $CdSO_4$ where purchased from Sigma Aldrich, and were used without any further purification.

### 2.1.3. ICP-OES Standard Solutions

For the preparation of the elemental standards for ICP-OES analysis, the following standards were used: 9998 mg/L S in water (Sigma Aldrich, München, Germany), 10,000 mg/L Al in 2 mol/L $HNO_3$ (Bernd Kraft, Duisburg, Germany), and 10,000 mg/L Cd in 2 mol/L $HNO_3$ (Bernd Kraft, Duisburg, Germany).

### 2.1.4. Ultrapure Water

As the medium for the adsorption, reaction and ICP-OES standards, ultrapure water purified using a Milli-Q Advantage A10® system (Millipore, Darmstadt, Germany) (TOC 5 ppb, resistivity of 18.2 MΩ·cm at 25 °C) was used.

### 2.2. *Synthesis of Hexanoyl-Modified Chitosan (H-Chitosan)*

In total, 1.3 g of the purchased chitosan (Ch90/60/A1 or Ch90/200/A1) was dissolved for 16 h in 175 mL 0.12 M acetic acid. The pH value of the solution was adjusted to 7.0 with 2 M NaOH, creating a slurry of partially precipitated chitosan. Hexanoyl chloride was added to the mixture dropwise according to the mass ratio (Table 1), and was then stirred at 600 rpm for 16 h at room temperature (r.t.) (Scheme S1). For the weight ratios 1:8 and 1:12, the hexanoyl chloride was added stepwise in 4.6 mL portions. After each injection, the mixture was stirred at r.t. for 16 h and then neutralized before adding the next 4.6 mL hexanoyl chloride. This approach was chosen because adding the entirety of the hexanoyl chloride at once leads to two phases and inhomogeneous products.

**Table 1.** Overview of the reaction parameters which were varied in order to obtain the different types of modified chitosan samples (H-chitosan).

| Chitosan | Hexanoyl Chloride Added in mL | Hexanoyl Chloride Added in mmol | (*w/w*) Ratio: Chitosan/Hexanoyl Chloride | Sample Code |
|---|---|---|---|---|
| Ch90/60/A1 (1.3 g) | 2.3 | 16.5 | 1:2 | H-Ch60-2 |
| | 9.2 | 65.8 | 1:8 | H-Ch60-8 |
| | 13.8 | 98.7 | 1:12 | H-Ch60-12 |
| Ch90/200/A1 (1.3 g) | 2.3 | 16.5 | 1:2 | H-Ch200-2 |
| | 9.2 | 65.8 | 1:8 | H-Ch200-8 |
| | 13.8 | 98.7 | 1:12 | H-Ch200-12 |

Afterwards, the pH value of the mixture was again adjusted to pH 7.0 using 2 M NaOH. The resulting solution (about 300 mL) was then precipitated in 2 L acetone, leading to a gelation of H-chitosan. The formed gel was separated through filtration using reduced pressure. Furthermore, it was thoroughly washed with hot methanol (about 60 °C) to remove residual hexanoyl chloride. The resulting gel was filled into 2–3 different small beakers (50 mL) and dried at 50 °C for 24 h.

### 2.3. *Adsorption Experiments with Heavy Metal Salts*

Stock solutions of each heavy metal salt were prepared with ultrapure water in a 1 L volumetric flask. The respective concentrations for the screening studies were 2 mg/L and 20 mg/L for $Al_2(SO_4)_3$, 0.03 mg/L and 0.3 mg/L for $CdSO_4$. The isotherms were measured from batch adsorption at the concentrations of 15, 30, 50, 60, 80, 90, 100, 120, 140, 200, 280, 320, 370, 400, 450, 500, 1000, and 1500 mg/L for $Al_2(SO_4)_3$, and 0.08, 0.8, 8.0, 20, 40, 85, 175, 250, 330, 420, 580, 625, 705, 930, 1090, 1275 and 1420 mg/L for $CdSO_4$. The pH of the solutions was not adjusted.

In total, 100 mg of the adsorbent was placed in a 50 mL centrifuge tube. Then, 30 mL of the heavy metal ion solution was added. After stirring for 24 h with a magnetic stirrer at 400 rpm at r.t., the samples were centrifuged for 8 min at r.t. with 11,000 rpm. Next, 8 mL of the supernatant was offset with 2 mL 20 wt.% nitric acid for ICP-OES analysis. In addition, the pH of the supernatant was measured.

### 2.4. Characterization and Analysis

2.4.1. Fourier-Transform Infrared Spectroscopy (FTIR)

Thin films of either native chitosan or the H-chitosans, respectively, were cast onto Ge internal reflection elements (IRE, 50 × 20 × 2 mm, Komlas GmbH, Berlin, Germany) from their respective solutions (50 µL) in 0.001 M HCl, dried at 50 °C, placed in a dedicated in-situ cell (IPF construction, M.M.) and purged using an $N_2$ stream. This in-situ cell was located in the SBSR (i.e., single-beam sample reference) ATR-FTIR attachment (Optispec, Neerach, Switzerland) integrated in the external sample compartment of the Tensor II FTIR Spectrometer (Bruker Optics GmbH, Ettlingen, Germany) and operated by dedicated macro (Optispec, Neerach, Switzerland) running within FTIR spectroscopy software (OPUS 7.0, Bruker Corporation, Billerica, MA, USA). Due to solubility issues, the samples H-Ch60-12 and H-Ch200-12 could not be measured as solid films.

Generally, FTIR spectra based on a resolution of 2 cm$^{-1}$ and 100 scans were recorded.

2.4.2. Particle Charge Detection (PCD) and Zetapotential vs. pH

By stirring the chitosans in 2% acetic acid for 2 h, a 0.05 g/L solution was prepared. The pH of around 3 was then adjusted to pH 1 with 1 M HCl. These solutions were stirred for 24 h at 50 °C. Subsequently, the solutions were treated for 2 h in an ultrasonic bath. In total, 10 mL of each solution was titrated against NaPES (0.001 M) in a particle charge detector, MÜTEK PCD-04, from the company BTG Instruments GmbH in Wessling, Germany. The streaming potential vs. pH was measured (MÜTEK PCD-04) by the titration of the solutions against a 0.1 g/L NaOH solution from their initial pH after dissolution up to a pH of 10.

2.4.3. Thermogravimetric Analysis (TGA)

For TGA measurements, the 1 Star System device from Mettler Toledo, Gießen, Germany was used. Approximately 5 to 8 mg of the sample was placed in a platinum crucible. The investigated temperature range was from 30 °C to 1000 °C, with a heating rate of 10 °C/min, under an air atmosphere at a flow rate of 40 mL/min.

2.4.4. Nitrogen Sorption Experiments

The nitrogen sorption experiments were conducted using the Autosorb iQ MP from Quantachrome Instruments, Boynton Beach, USA. Preliminary, the samples were dried in a vacuum oven at 100 °C for at least 20 h. Furthermore, they were degassed at 90 °C with ultra-high vacuum (5 × 10$^{-10}$ mbar) for another 20 h. The measurements were performed at 77 K with about 100 mg of the sample. The resulting data were fitted using the Brunauer Emmett Teller method (BET, fitting range between 0.01–0.3 p/p$_0$).

2.4.5. Scanning Electron Microscope (SEM) and Energy-Dispersive X-ray Spectroscopy (SEM-EDX)

For the SEM imaging of the modified chitosan on a nanometric scale, an Ultra plus (Carl Zeiss Microscopy GmbH, Oberkochen, Germany) with a secondary electron detector (SE2) was used. Before the analysis, the samples were attached to an aluminium pin sample tray with double-sided adhesive carbon tape and coated with 3 nm platinum in order to avoid electrostatic charging. SEM-EDX for the elemental mapping of the samples after the adsorption process was carried out with the same instrument, for which the detector was switched to an energy-dispersive X-ray spectroscopy detector (X-Flash 5060F, thermoelectrically cooled (LN2free), 4 × 10 mm$^2$ detector active area). The samples were fixed on a double-sided adhesive carbon tape on an aluminum pin sample tray. The measurements were carried out in high vacuum mode with an acceleration voltage of 6 keV at different magnifications.

### 2.4.6. Elemental Analysis

For the elemental analysis, a vario MICRO cube from the company Elementar, Langenselbold Germany was used.

### 2.4.7. Nuclear Magnetic Resonance (NMR)

The chitosan samples were dissolved in $D_2O$ by adding DCl (0.1 M). The $^1H$ NMR spectra were measured using an Avance III 500 MHz Bruker Biospin system (Bruker Corporation, Billerica, MA, USA).

### 2.4.8. Inductive Coupled Plasma-Optical Emission Spectroscopy (ICP-OES)

The ICP-OES measurements were conducted with the iCAP 7400 from Thermo Scientific (Waltham, MA, USA) to determine the heavy metal ion concentration in the simulated water before and after the adsorption process. Thus, five Standards in 4 wt.% aqueous $HNO_3$ were used: standard 1 (Cd and S, each 1 mg/L), standard 2 (Al and S, each 2 mg/L), standard 3 (Al and S, each 20 mg/L), standard 4 (Cd and S, each 100 mg/L), and standard 5 (Al, Cd, and S, each 2000 mg/L). For the calibration, each standard was additionally measured in dilutions of 1:2, 1:4 and 1:8. To each sample (8 mL) was added 2 mL 20% nitric acid prior to the analysis.

### 2.4.9. pH-Dependent Solubility Measurements

For the solubility measurements, the pH of 30 mL ultrapure water was adjusted with HCl. In total, 100 mg of the respective H-Chitosan was added and stirred for 24 h. Subsequently, the sample was filtered, and the residual mass was determined after drying.

### *2.5. Calculation of the N-Acylation Degree*

In order to improve the accuracy of the determination of the N-acylation substitution degree, two different methods were used.

Primarily, $^1H$-NMR measurements at 90 °C in DCl 0.1 M were used to determine the DS%$_{NMR}$. In order to increase the accuracy, we used the integrals of four groups of the hexanoyl functionality at 0.8 ppm (-CH$_3$), 1.3 ppm (two -CH$_2$), and 1.5 ppm (-CH$_2$). By dividing through the number of contributing H-atoms (9) and normalizing to H-1 (consistent of I$_{H-1\alpha}$ + I$_{H-1\beta}$), the DS%$_{NMR}$ is calculated:

$$DS\%_{NMR} = \frac{I_{0.8\ ppm} + I_{1.3\ ppm} + I_{1.5\ ppm}}{9 \times \left(I_{H-1\alpha} + I_{H-1\beta}\right)} \times 100\% \tag{1}$$

The second is based on FTIR measurements, and uses a modified version of the method proposed by Moore and Roberts [26], which was recently reported [24]. Based on the ratio of the integral of the Amide I band at 1656 cm$^{-1}$ and the intense n(C-O) band (ether and hydroxyl groups) diagnostic for polysaccharides (SACC) at 1100 cm$^{-1}$ for H-chitosan (denoted H-Ch) and native chitosan (denoted N-Ch), the DS%$_{FTIR}$ is calculated. In this way, the known degree of acetylation DS%$_{Acetyl}$ of the native chitosan is used as a reference. For the proper determination of the Amide I band, integral line shape analysis based on 5 components (Gaussian/Lorentzian) representing the spectral range between 1700 and 1475 cm$^{-1}$ was applied as published similarly [24]. Exemplary original and fitted curves and underlying components are given in the Supplementary Materials (Figures S3.1 and S4.1). The component at 1656 cm$^{-1}$ was assigned to the Amide I band, and its integral was assigned to either I$_{Amide\ I\ H-Ch}$ or I$_{Amide\ I\ N-Ch}$. The integral of the n(C-O) band was determined by integration between 1200 and 950 cm$^{-1}$, and was assigned to either I$_{SACC\ H-Ch}$ or I$_{SACC\ N-Ch}$ in Equation (2):

$$DS\%_{FTIR} = \left[\left(\frac{I_{Amide\ I\ H-Ch}}{I_{SACC\ H-Ch}}\right) \times \left(DS\%_{Acetyl} \times \frac{I_{SACC\ N-Ch}}{I_{Amide\ I\ N-Ch}}\right) - DS\%_{Acetyl}\right] \times 100\% \tag{2}$$

### 2.6. Theoretical Model

In order to determine the sorption efficiency of the samples, the concentration of the adsorbed metal ions and sulfate ions in equilibrium were detected by ICP-OES and used in Equation (3), wherein the calculation of the sorption as a percentage can be seen, with $c_0$ as the concentration of the respective ion in the initial solution and $c_{eq}$ as the concentration after reaching equilibrium.

$$\text{adsorption efficiency} = 100\% \times \frac{c_0 - c_{eq}}{c_0} \tag{3}$$

The respective sorption capacity q in equilibrium was calculated as follows:

$$q_{eq} = \frac{(c_0 - c_{eq}) \times V_L}{m_A} \tag{4}$$

$V_L$ refers to the given volume of the adsorptive solution, and $m_A$ refers to the mass of the sorbent material used in the experiment.

Different fitting models were used to model the sorption process, as were Langmuir Equation (5) [27], Sips Equation (6) [28] and Dubinin-Radushkevich Equations (7)–(10) [29,30] isotherm models in non-linear form.

The Langmuir model is as follows:

$$q_{eq} = \frac{Q_m \times K_L \times c_{eq}}{1 + K_L \times c_{eq}} \tag{5}$$

The Sips model is as follows:

$$q_{eq} = \frac{Q_m \times K_S \times c_{eq}^n}{1 + K_S \times c_{eq}^n} \tag{6}$$

$K_L$ and $K_S$ thereby represent the equilibrium constants of the Langmuir and Sips models, respectively, $Q_m$ is the maximum adsorption capacity, and n is the Sips model exponent.

The Dubinin-Radushkevich model is as follows:

$$q_{eq} = Q_m \times \exp\left(-\beta_{DR} \times \varepsilon^2\right) \tag{7}$$

$$\varepsilon = RT \times \ln\left(\frac{c_S}{c_{eq}}\right) \tag{8}$$

$c_S$ represents the solubility of the adsorbate. The term inside the logarithm $\frac{c_S}{c_{eq}}$ can be exchanged for $1 + \frac{1}{c_{eq}}$ according to Zhou, leading to the same numerical solution with two requirements. First, this is only possible for values of $c_{eq} \ll c_S$. Second, it is important to use molar concentrations for the fitting [31]. We have implemented both as suggested. This leads to the following term for the Polanyi potential:

$$\varepsilon = RT \times \ln\left(1 + \frac{1}{c_{eq}}\right) \tag{9}$$

$\beta_{DR}$, as the activity coefficient, is related via Equation (10) to the mean free energy of adsorption $E_{ads,DR}$ [32,33].

$$E_{ads,DR} = \frac{1}{\sqrt{2 \times \beta_{DR}}} \tag{10}$$

From the Langmuir model, a thermodynamic evaluation of the adsorption process can be made. Here, the Langmuir equilibrium constant is related to the change in the Gibbs free energy of adsorption $\Delta G°$ via Equation (11):

$$\Delta G° = -RT \times \ln(K_a) \tag{11}$$

$K_a$ represents the dimensionless activity coefficient, which is exchanged by the Langmuir equilibrium constant in L/mol, deliberately ignoring the mismatched dimensions [34].

## 3. Results and Discussion

### 3.1. Characterization of H-Chitosan

In order to investigate the influence of the hexanoyl chloride modification on the adsorption behavior of chitosan, we synthesized six different types of hexanoyl-chitosan (H-chitosan). In this way, the impact of the molar mass of the chitosan and the amount of hexanoyl chloride on the substitution reaction was examined. Hence, we used Ch90/200/A1 and Ch90/60/A1 for the modification, as both chitosans featured a deacetylation degree of 90% and an ash content of 1% (A1 stands for a 1% ash content). The values 200 mPas and 60 mPas stand for the viscosity, which is directly proportional to the molar mass of the chitosan. Both investigated chitosan samples possessed a relatively low molar mass, where Ch90/60/A1 and Ch90/200/A1 featured a molar mass of 90–150 kDa and roughly 620 kDa, respectively. Furthermore, the amount of hexanoyl chloride added to the chitosan was varied in order to obtain different substitution degrees. Hence, the six synthesized samples were labeled according to the following scheme: H-Ch60-2, where H stands for hexanoyl chloride, Ch60 stands for chitosan and the viscosity of the chitosan, and 2 stands for the mass ratio of hexanoyl chloride/chitosan used in the synthesis (see Table 1). The synthesis is described in detail in the experimental section (see Section 2.2). In short, hexanoyl chloride was added to a chitosan solution with a pH of 7.0, and was stirred for 16 h. Subsequently, the formed gel was filtered, washed with hot methanol (around 60 °C), and dried. The modified polymers were characterized by proton nuclear magnetic resonance ($^1$H-NMR) spectroscopy, Fourier transform infrared (FTIR) spectroscopy, thermogravimetric analysis (TGA), zetapotential measurements, and nitrogen sorption in order to confirm the structure, calculate the substitution degree, and investigate the physicochemical properties.

In order to investigate the structure and to calculate the substitution degree, $^1$H-NMR and FTIR measurements were conducted (see Figures 1a and 1c, respectively). The $^1$H- NMR spectra of the native chitosan (Ch90/60/A1 and Ch90/200/A1) and the complementary H-chitosan samples substituted with a 1:12 ratio of hexanoyl chloride (H-Ch60-8 and H-Ch200-8) are shown in Figure 1a. The signals at 4.89 ppm and 4.52–4.67 ppm are ascribed to H-1$^b$ and H-1$^a$ protons connected to the anomeric center of the glycosidic bond. The further chitosan-related signals are assigned as follows: 3.50–4.00 ppm (H-2 to 6), 3.12–3.28 ppm (H-2) and 2.02 ppm (H-7). These peaks were identical in all spectra (compare Figures S1 and S2). The spectra of the modified chitosan samples (here H-Ch60-12 and H-Ch200-12) featured several additional peaks in the aliphatic region, which can be attributed to the hexanoyl residue. The peaks at 2.26 ppm (H-8), 1.54 ppm (H-9) and 1.27 ppm (H-10 and 11) originate from the different $CH_2$ functionalities, whilst the peak at 0.83 ppm (H-12) derives from the terminal $CH_3$ group. In Figures S1 and S2, the correlation of the integrals of the hexanoyl related signals (normalized to H-1$^b$ + H-1$^a$ = 1.0) with the increasing amount of modification agent is shown for the 60 mPas and the 200 mPas H-chitosan, respectively. The chitosans with higher levels of substitution exhibited a loss of spin multiplicity of the hexanoyl-related signals due to micelle formation [35].

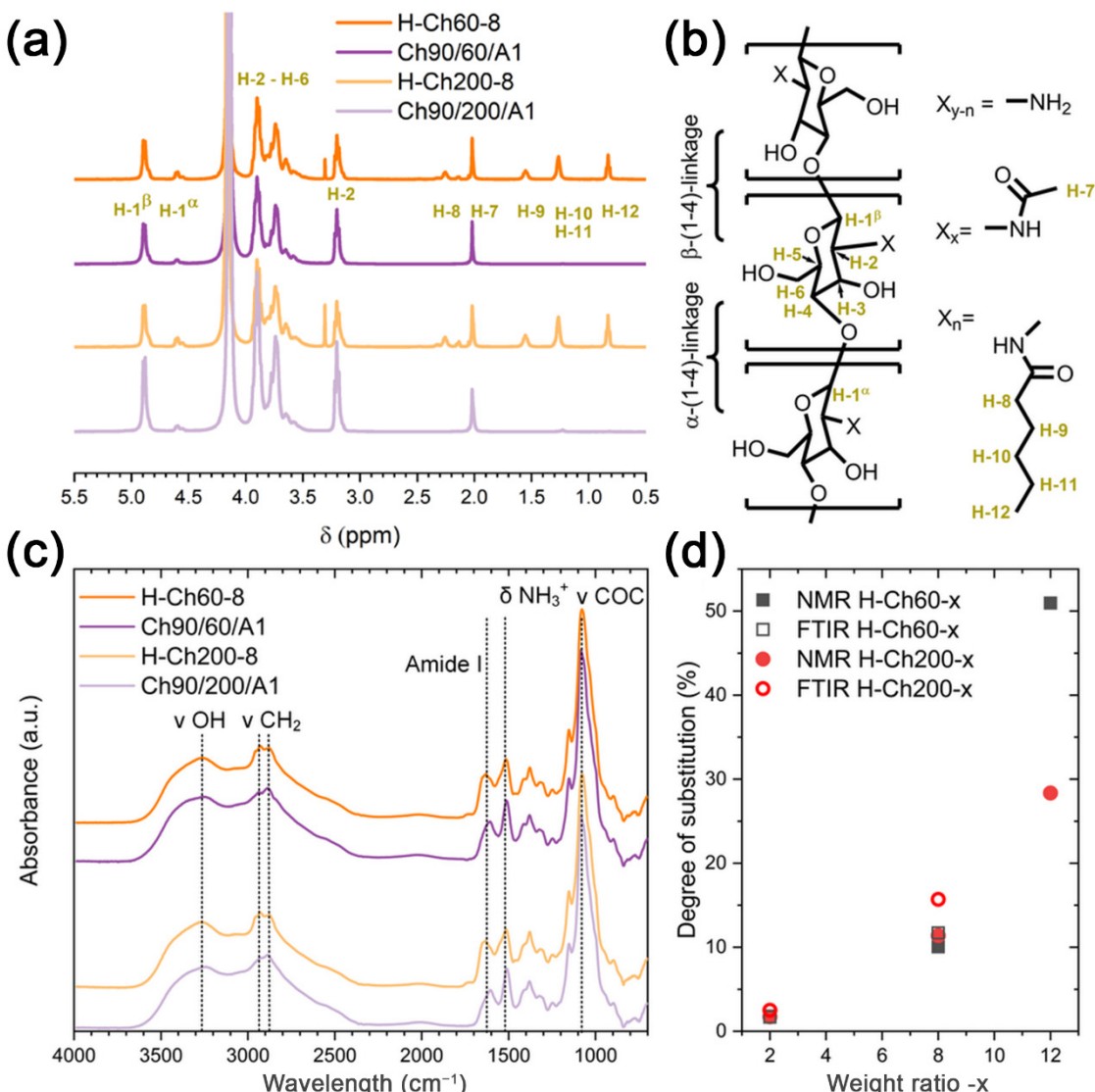

**Figure 1.** (**a**) [1]H-NMR spectra of Ch90/60/A1 (dark purple), H-Ch60-8 (orange), Ch90/200/A1 (light purple), and H-Ch200-8 (yellow) with peak assignments, and (**b**) the structure with proton-to-peak assignment (all of the spectra in Figures S1 and S2). (**c**) ATR-FTIR spectra of H-Ch60-8 (orange) compared to Ch90/60/A1 (dark purple), and H-Ch200-8 (yellow) compared to Ch90/200/A1 (light purple), with band assignments at 3600–3100 cm$^{-1}$ ($\nu$(OH)), 2918 cm$^{-1}$ and 2879 cm$^{-1}$ ($\nu$(CH$_2$)), 1653 cm$^{-1}$ (Amide I, increases with N-acylation), 1510 cm$^{-1}$ ($\delta$(NH$_3^+$)), and 1080 cm$^{-1}$ (polysaccharide, $\nu$ C-O-C and $\nu$ C-OH) (all of the spectra in Figures S3 and S4). (**d**) Comparison of the degree of substitution in the dependency of the applied weight ratio of hexanoyl chloride to chitosan (x). H-Ch60-12 and H-Ch200-12 were not measured due to solubility issues.

Furthermore, the formation of the side product hexanoic acid (through hydrolysis) and residual hexanoyl chloride cannot be precluded via the spin multiplicity for the higher-substituted H-chitosan samples (weight ratios 1:8 and 1:12). Therefore, further investigations via FTIR spectroscopy were conducted (see Figure 1c, Figures S3 and S4).

The hydroxyl functionalities of the glucosamine structure, together with water residues, are the origin of the broad $\nu$(OH) band between 3600 and 3100 cm$^{-1}$ in the FTIR spectra (see Figure 1c). The CH$_2$ groups of both the glucosamine structure and the hexanoyl modification are attributed to the $\nu$(CH$_2$) at 2918 cm$^{-1}$ and 2879 cm$^{-1}$. The Amide I band at around 1650 cm$^{-1}$ confirms the presence of amide functionalities in the spectra of the H-chitosan and the non-modified chitosan. At 1510 cm$^{-1}$ the $\delta$(NH$_3^+$) band is located,

resulting from the protonation of the amino functionalities when casting the films from HCl solutions. The most prominent band in all of the spectra is the $\nu$(C-O) band at 1080 cm$^{-1}$, which originates from C-O-C and C-OH linkages within the polysaccharide backbone.

Common ester bands at 1700 cm$^{-1}$, 1200 cm$^{-1}$ and 1100 cm$^{-1}$ are not displayed in the IR spectra, indicating that only the amino functionalities were subjected to the modification [36]. Additionally, no bands related to acyl chloride or carboxylic acids were observed in any of the spectra. Hence, neither hexanoyl chloride nor the hydrolysis product hexanoic acid are contained within the modified chitosan material.

Both methods, NMR and FTIR, can be used to calculate the DS%, as we recently reported [24] (compare Section 2.5., Equation (1). An overview for the different DS% of the H-chitosan samples is provided in Table 2.

**Table 2.** Degree of substitution of the H-chitosan samples determined via NMR and FTIR, respectively [24]. Due to aggregation and solubility issues, H-Ch60-12 and H-Ch200-12 were not determinable with the FTIR method.

| Sample | Weight Ratio Chitosan: Hexanoyl Chloride | DS % NMR | DS % FTIR |
|---|---|---|---|
| H-Ch60-2 | 1:2 | 1.8% | 1.7% |
| H-Ch60-8 | 1:8 | 10.1% | 11.6% |
| H-Ch60-12 | 1:12 | 51.1% | - |
| H-Ch200-2 | 1:2 | 1.7% | 2.5% |
| H-Ch200-8 | 1:8 | 11.4% | 15.7% |
| H-Ch200-12 | 1:12 | 28.4% | - |

The NMR- and the FTIR-based methods show a significant increase in the DS% for higher weight ratios of hexanoyl chloride, which does not scale linearly to the applied amount of hexanoyl chloride (Figure 1d). During the synthesis, the higher amounts of hexanoyl chloride in the reaction mixture resulted in a stronger pH change of the reaction mixture in comparison to lower amounts of hexanoyl chloride, such as the 1:2 ratio. Because the reaction is highly pH dependent [24], the drastic decrease in the pH value for higher amounts of hexanoyl chloride resulted in the phase separation of the reaction mixture. In order to enable a complete reaction with the total amount of hexanoyl chloride added to the reaction mixture, a neutralization between the stepwise injection of hexanoyl chloride (only for weight ratios of 1:8 and 1:12) was conducted. Unreacted hexanoyl chloride from the first injection may further contribute to the substitution, instead of being discarded as for the lower weight ratios of 1:2. Therefore, not only does the higher amount of hexanoyl chloride contribute to the high DS% but the extra neutralizations also have positive influence. Additionally, the already-attached hexanoyl moieties may have a directing effect on non-reacted hexanoyl chloride. Furthermore, the Ch90/60/A1 converted with 12 mass equivalents of hexanoyl chloride exhibited a significantly higher DS% (approximately twice) than the matching reaction with Ch90/200/A1. This is due to the higher mobility of the shorter polymer chains, which is particularly strong when additional hydrophobic interactions counteract an efficient solution process.

TGA measurements were performed in order to investigate the impact on thermal stability caused by the hexanoyl modification (Figure S5). Whilst the pure chitosans start to decompose at around 256 °C, the H-chitosans show a bimodal or even trimodal weight loss profile. The first step begins at around 277 °C for the lower-substituted H-chitosans, and decreases to around 250 °C with the increasing substitution degree, which is in accordance to the findings of Peesan et al. [37]. The second decomposition step begins at 459 °C for weight ratios of 1:2 and 1:12, which also show an additional loss at around 540 °C. The 1:8 ratio shows a bimodal decomposition, with the second step starting at higher values of around 525 °C.

For the adsorption processes, the specific surface area (SSA) and the potential pore structure are relevant parameters. Thus, N$_2$ sorption analysis was conducted. Neither the

pure chitosan nor the H-chitosan samples showed a significant SSA (see Figures S6 and S7) larger than 5 m$^2$/g in the dry state. Thus, no micro- or mesopores were found. The specific surface area measured can be attributed to macroporosity or the outer surface area only. Thus, the exhibited specific surface area is insignificant when compared to the swelling in water. Here, chitosan—as a linear polymer—is known to swell in water to a relatively large extent [38], resulting in a material suitable for water treatment in aqueous solution.

Furthermore, in order to investigate the macroscopic surface structure, images were taken via SEM and SEM-EDX measurements (see Figures S8 and S9). These show heterogeneous particles with structured surfaces. Larger pores such as macropores were not observed on the SEM images. Furthermore, the elements were homogenously distributed over the sample, indicating that no hydrophobic domains were formed during the drying process.

In order to investigate the impact of the varying ratios of hexanoyl on the surface charge, streaming potential and charge density measurements were conducted. For a thorough analysis, however, the samples had to be dissolved first. Because the solubility of the chitosan was reduced by the modification with hexanoyl chloride, an improved protocol for dissolution, including ultrasonic treatment and heating, was applied [24]. This reduced the larger, non-dissolved fragments to suspended particles (micelles and aggregates), which were monitored via turbidity and dynamic light scattering measurements (DLS). In Figure 2a,b, the number distributions are shown for all of the samples (the intensity and volume distribution are shown in Figures S10 and S11). For Ch90/60/A1 and the corresponding modified derivatives, larger particles were observed for the samples H-Ch60-8 and H-Ch60-12. Hence, the higher DS% leads to aggregation, which was also observed in the turbidity measurements. In this way, the H-Ch60-12 exhibited smaller particles than H-Ch60-8 despite the higher DS%. This may be due to the larger attraction within the hydrophobic moieties. The molecular weight of the modified chitosan seems to influence this effect as well, as Ch90/200/A1 does not show this effect. On the one hand, the particle size of H-Ch200-12 was larger than that for H-Ch200-8. The H-Ch200-2 particles, on the other hand, are smaller than the native, swollen chitosan polymer. The attracting forces of the hydrophobic moieties may cause this phenomenon, leading to a decrease in particle size for low DS%, which is then increased with higher DS% due to enhanced aggregation. When a threshold is surpassed, the DS% influence on the density of the aggregates seems to outweigh the influence on the general tendency of aggregation, resulting in smaller particles.

The streaming potential vs. pH curves for all of the samples featured similar curve progressions and an isoelectric point (IEP) at approximately pH 7 (Figure 2d,e, compare Section 2.5 and Table S2).

In order to determine the number of accessible amino functionalities, which are crucial for heavy metal and oxyanion adsorption, colloidal titration was performed. In this way, for both chitosans a strong decrease of the charge density was observed with the increasing DS% (Figure 2f). This is due to the reduced number of protonizable amino functionalities caused by the amide formation. Further, the accessibility of the remaining amino functionalities is diminished through micelle and aggregate formation, which is examined by an increase in turbidity (Figure 2c).

Solubility studies in acidic media were conducted for Ch90/60/A1 and its derivatives, as Ch90/60/A1 showed an overall better substitution (Figure 3). H-Ch60-2 exhibited an overall increased pH stability when compared with Ch90/60/A1. H-Ch60-8 and H-Ch60-12 showed a higher residual mass at pH 1 and 2 than the unmodified chitosan but similar or less at pH 3 and 4. This can be a consequence of the method used for determination (Section 2.5). Both of these samples exhibited a high tendency for micelle formation, as proven in the turbidity and DLS measurements (Figure 2). Therefore, a great portion of these modified chitosans may actually not be dissolved into individual polymer chains but in a micellar state small enough to pass through the filter. Because the objective of the modification was not only to reduce the solubility of the chitosan but also to achieve a

good separability from the adsorption media by filtration, H-Ch60-2 exhibited the most preferable characteristics. The advantage is that H-Ch60-2 also has the lowest DS%, leaving more amino functionalities for the adsorption process.

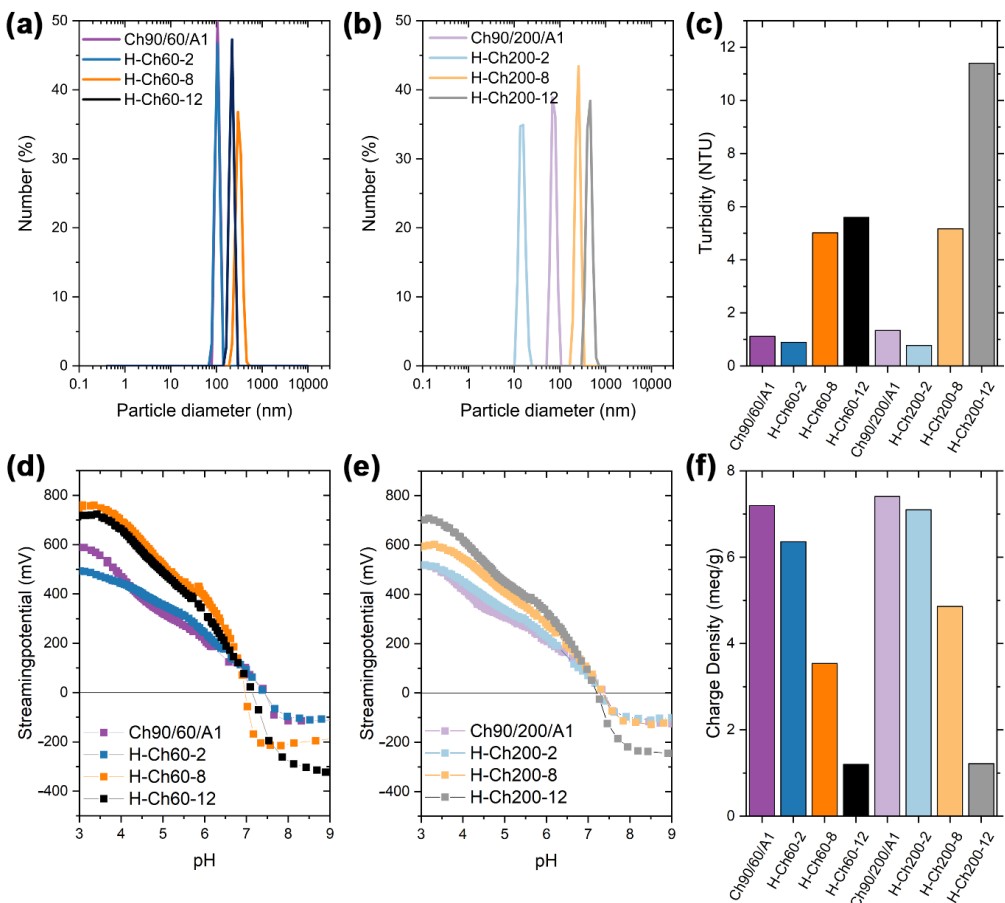

**Figure 2.** (**a**,**b**) DLS, (**c**) turbidity, (**d**,**e**) streaming potential vs. pH, and (**f**) charge density measurements of Ch90/60/A1 and Ch90/200/A1, and the respective hexanoyl-modified derivatives.

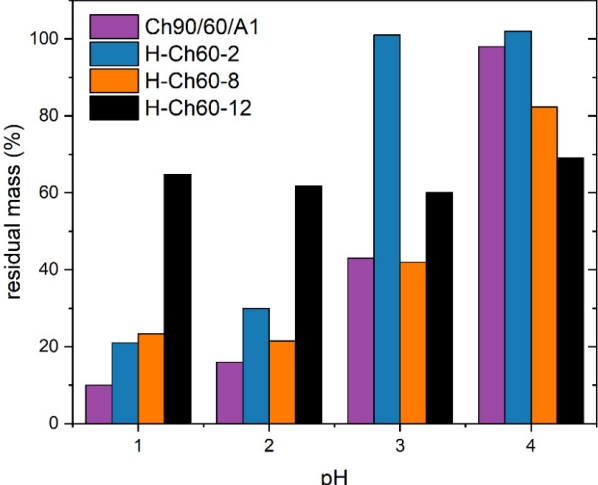

**Figure 3.** Residual mass of Ch90/60/A1 (purple), H-Ch60-2 (blue), H-Ch60-8 (orange) and H-Ch60-12 (black) after dissolution at the respective pH for 24 h.

### 3.2. Adsorption of Metal Ions

The modified H-chitosan samples, as well as the pure chitosan, were tested for the removal of $Al^{3+}$, $Cd^{2+}$, and $SO_4^{2-}$. In order to compare the adsorption properties of these eight samples, two initial concentrations corresponding to the 10- and 100-fold of the WHO guideline values for metal ions in drinking water were tested. Here, for $Al^{3+}$ and $Cd^{2+}$, the suggested maximum concentrations are 0.2 mg/L and 0.003 mg/L, respectively [39]. The adsorption rates obtained for the respective metal ions and sulfate in batch experiments are presented in Figure 4 (the respective pH values of the experiments are displayed in Figures S12–S15). For sorption experiments with $CdSO_4$ solution, the monitoring of $SO_4^{2-}$ was not possible due to the low concentration range.

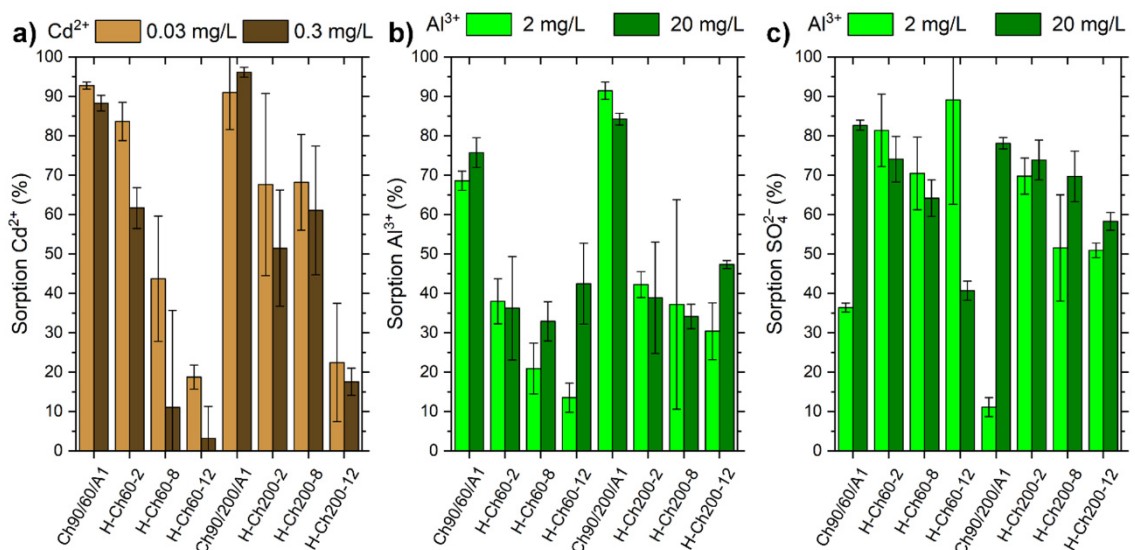

**Figure 4.** Results for the adsorption experiments with either $CdSO_4$ or $Al_2(SO_4)_3$ solution on Ch90/60/A1, Ch90/200/A1, and their respective modifications with different weight ratios of hexanoyl chloride. Depicted is the percentage removal of (**a**) $Cd^{2+}$, (**b**) $Al^{3+}$, and (**c**) $SO_4^{2-}$ ions. Here, the given concentration corresponds to the concentration of the respective metal ion. The corresponding pH values for the experiments can be seen at Figures S12–S15.

In general, an increasing weight ratio of hexanoyl chloride yields a decrease in the $Al^{3+}$ and $Cd^{2+}$ adsorption ratio (see Figure 4). This effect can be attributed to (a) less amines being available for the coordination of $Al^{3+}$ and $Cd^{2+}$ due to the substitution, and (b) the blocking of amines by micelle formation caused by hydrophobic interactions. Additionally, H-Ch200-8 and H-Ch200-12 featured approximately better adsorption rates compared to the respective H-Ch60-8 and H-Ch60-12 samples with a lower molar mass. Here, the higher molecular weight potentially facilitates chelation, increasing the adsorption in the low-concentration region. H-Ch60-2 is an exception to this behavior, with the highest achieved adsorption rates for $Cd^{2+}$ among the modified chitosan samples, as the DS% is low enough to still enable strong interaction towards the adsorbed species.

Comparing the achieved adsorption rates of $Al^{3+}$ with the corresponding anion sulfate, the adsorption rate of sulfate is often nearly twice as high as the adsorption rate of $Al^{3+}$. This might be explained by the relatively low initial pH value of the $Al_2(SO_4)_3$ solution leading to a positively charged surface area, which prefers the ionic interaction with the negatively charged sulfate ion (see Figure 2d,e and Figure S12 and S13). Furthermore, the differences in the adsorption rates in the dependence of the hexanoyl chloride yield as well as for the molar mass were relatively minor for $Al^{3+}$ and sulfate in comparison to $Cd^{2+}$.

Although the adsorption rates of the pure chitosan Ch90/60/A1 and Ch90/200/A1 featured higher values for the two investigated metal ions than the modified chitosan samples did, the adsorption rates for sulfate were comparable to or lower than those

of the H-Ch200-2 and H-Ch60-2 samples. In order to further investigate the adsorption process of $Al^{3+}$, $Cd^{2+}$ and sulfate, H-Ch60-2 featuring the best removal for $Cd^{2+}$ and very good removal rates for $Al^{3+}$ was selected for the concentration dependent batch adsorption. Furthermore, H-Ch60-2 requires the lowest amount of hexanoyl chloride for modification process.

Therefore, adsorption isotherms for $Al_2(SO_4)_{3(aq)}$ and $CdSO_{4(aq)}$ were conducted and subsequently modeled with Langmuir, Sips and Dubinin–Radushkevich isotherm models (described in Section 2.6). After the adsorption experiments, the respective samples were washed with ultrapure water, dried, and comprehensively analyzed by SEM-EDX, in order to investigate the adsorption mechanism.

Although it showed higher adsorption efficiency in the previous experiments, adsorption experiments with pure chitosan are not practical for higher $c_0$ values due to the acidic pH, which consequently leads to the dissolution of the chitosan, as shown in Figure 3. For that reason, only adsorption experiments with the hexanoyl-modified chitosan were conducted.

The adsorption capacity of H-Ch60-2 for $Al_2(SO_4)_{3(aq)}$ and the elemental distribution of Al and S are given in Figure 5. The experimental values for the adsorption capacities were approximately 2.3 mmol/g for sulfate ions and 1.6 mmol/g for $Al^{3+}$. A higher amount of sulfate ions was adsorbed due to the acidic $pH_0$ between 4.3 and 3.9 resulting in the protonation of the amino groups (see Figure 2), thus allowing ionic interactions with sulfate. Another factor for the higher affinity toward sulfate is that $Al^{3+}$ is the predominant cationic species in this pH range [40]. Schwarz et al. [4] showed an increase in the adsorption capacity of metal ions if sulfate as a bivalent anion is present as the corresponding anion.

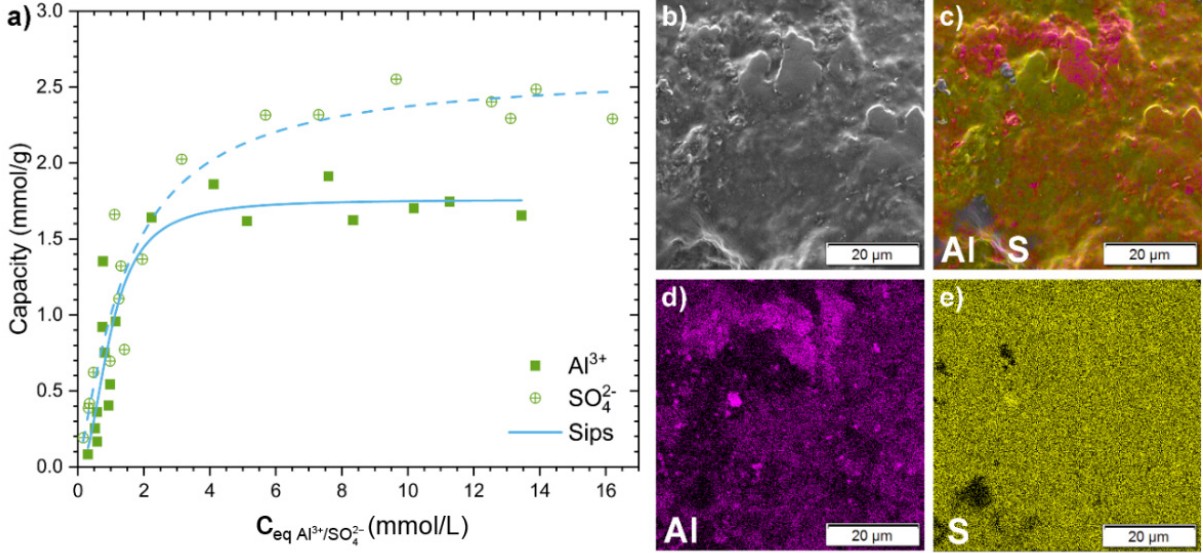

**Figure 5.** (**a**) Adsorption capacity for $Al^{3+}$ (filled squares) and $SO_4^{2-}$ (crossed circles) from $Al_2(SO_4)_3$ solutions at the respective equilibrium concentrations $c_{eq}$ with the corresponding Sips isotherm model fits (blue, solid line for $Al^{3+}$ and dashed line for $SO_4^{2-}$). The corresponding pH values for the experiments can be seen at Figure S16. (**b**–**e**) SEM-EDX image and mapping of H-Ch60-2 after the adsorption experiment with $Al_2(SO_4)_3$ solution ($c_0 = 370$ mg/L $Al^{3+}$). (**b**) SEM image of H-Ch60-2 after the adsorption with $Al_2(SO_4)_{3(aq)}$; (**c**–**e**) SEM-EDX elemental mapping of (**c**) Al (purple) and S (yellow), (**d**) Al (purple), and (**e**) S (yellow). The corresponding EDX spectra are shown in Figure S18. The SEM-EDX and SEM images for H-Ch60-2 before the adsorption process are shown in Figure S8.

From the chosen models, the adsorption process—both of $Al^{3+}$ and $SO_4^{2-}$ ions—can be best described with Sips isotherm equation (see $R^2$ and fitting parameters in Table 3). The calculated adsorption capacities $Q_m$ of 1.8 mmol/g for $Al^{3+}$ and 2.6 mmol/g for $SO_4^{2-}$ are in agreement with the experimental data, indicating that a maximum capacity was

achieved. As the Sips isotherm model is a combination of the Langmuir and Freundlich model, the model's exponent n refers to the heterogeneity of the isotherm. When this exponent is close to one, the Sips isotherm equation reduces to the Langmuir equation, thereby showing monolayer adsorption. In the modelling of $Al^{3+}$ and $SO_4^{2-}$, a strong difference for n between the two ions can be seen. For sulfate, n has a value of 1.2 and is closer to unity in comparison to the n of $Al^{3+}$ with Section 2.2.

**Table 3.** Fitting parameters for Langmuir, Sips and Dubinin–Radushkevich isotherm models for adsorption from $Al_2(SO_4)_3$ solution. Here, $Q_m$ is the maximal sorption capacity, K is the equilibrium constant for the respective model, n is the Sips isotherm model exponent, and $E_{ads,DR}$ is the mean free energy of adsorption derived from the Dubinin–Radushkevich model. $R^2$ corresponds to the coefficient of determination (COD).

| Ion | Model | $Q_m$ mmol/g | K * | $\beta_{DR}$ $10^{-9}$ $mol^2/J^2$ | n | $E_{ads,DR}$ kJ/mol | $\Delta G^\circ$ kJ/mol | $R^2$ (COD) |
|---|---|---|---|---|---|---|---|---|
| $Al^{3+}$ | Langmuir | $2.08 \pm 0.20$ | $0.65 \pm 0.20$ | – | – | – | –6.07 | 0.799 |
| | Sips | $1.76 \pm 0.12$ | $1.02 \pm 0.34$ | – | $2.18 \pm 0.79$ | – | – | 0.850 |
| | Dubinin-Radushkevich | $3.59 \pm 0.64$ | – | $5.17 \pm 0.95$ | – | $9.84 \pm 0.91$ | – | 0.734 |
| $SO_4^{2-}$ | Langmuir | $2.78 \pm 0.15$ | $0.58 \pm 0.11$ | – | – | – | –5.97 | 0.926 |
| | Sips | $2.60 \pm 0.21$ | $0.63 \pm 0.12$ | – | $1.22 \pm 0.28$ | – | – | 0.929 |
| | Dubinin-Radushkevich | $4.52 \pm 0.47$ | – | $5.05 \pm 0.6$ | – | $9.95 \pm 0.59$ | – | 0.884 |

* The unit of K for the Langmuir model is L/mmol, and for the Sips model it is $(L/mmol)^n$.

The mean free energy of adsorption ($E_{ads,DR}$) can be derived from the Dubinin–Radushkevich model, which corresponds to the energy needed to remove an adsorbed molecule from its adsorption site to an infinite distance. For both adsorbed species, the values are in the typical range for physisorption. Hence, H-Ch60-2 featured a slightly stronger binding toward $SO_4^{2-}$ ($E_{ads,DR}$ = 9.85 kJ/mol for $Al^{3+}$ vs. $E_{ads,DR}$ = 9.95 kJ/mol for $SO_4^{2-}$).

The negative values given for the change in free Gibbs energy $\Delta G^\circ$ show that the adsorption process is exergonic [34].

Additionally to the theoretical models, the H-Ch60-2 sample was further analyzed by SEM-EDX in order to acquire more detailed information about the adsorption mechanism. For the adsorption experiments with $Al_2(SO_4)_3$, the SEM image and elemental distribution from SEM-EDX are shown in Figure 6. According to the SEM-EDX mapping, the adsorption of $Al^{3+}$ occurred in a clustered fashion on the sample surface. In contrast, S was found to be homogenously distributed on the whole H-Ch60-2 sample surface. This indicates an adsorption of $Al^{3+}$ in these clusters as hydroxy or oxygen species onto the chitosan without significant interaction with $SO_4^{2-}$ in terms of cooperative adsorption mechanism.

Figure 6 shows the adsorption isotherm for $CdSO_4$ solution on H-Ch60-2. In contrast to $Al_2(SO_4)_3$, the adsorption of sulfate and cadmium ions onto the H-Ch60-2 occurred with adsorption capacities in an equimolar range. The experimental values possessed an adsorption capacity of approximately 2.0 mmol/g for $SO_4^{2-}$, and around 1.7 mmol/g for $Cd^{2+}$. As fitting models, Langmuir, Sips and Dubinin–Radushkevich isotherm models were chosen with fitting parameters displayed in Table 4. In terms of the $R^2$ values, the Sips model fit was best suited to model the adsorption behavior (see Table 4, Figure 6, Figures S19 and S20). The calculated maximum adsorption capacities from both the Sips and Dubinin–Radushkevich models were in the expected range of the plateau of the adsorption capacities, with 1.7 mmol/g and 2.1 mmol/g for $Cd^{2+}$ and $SO_4^{2-}$, respectively. Similarly to $Al_2(SO_4)_3$, $E_{ads,DR}$ is in the typical range for physisorption for both adsorbed species, and $\Delta G^\circ$ shows the spontaneity of the adsorption process with its negative values.

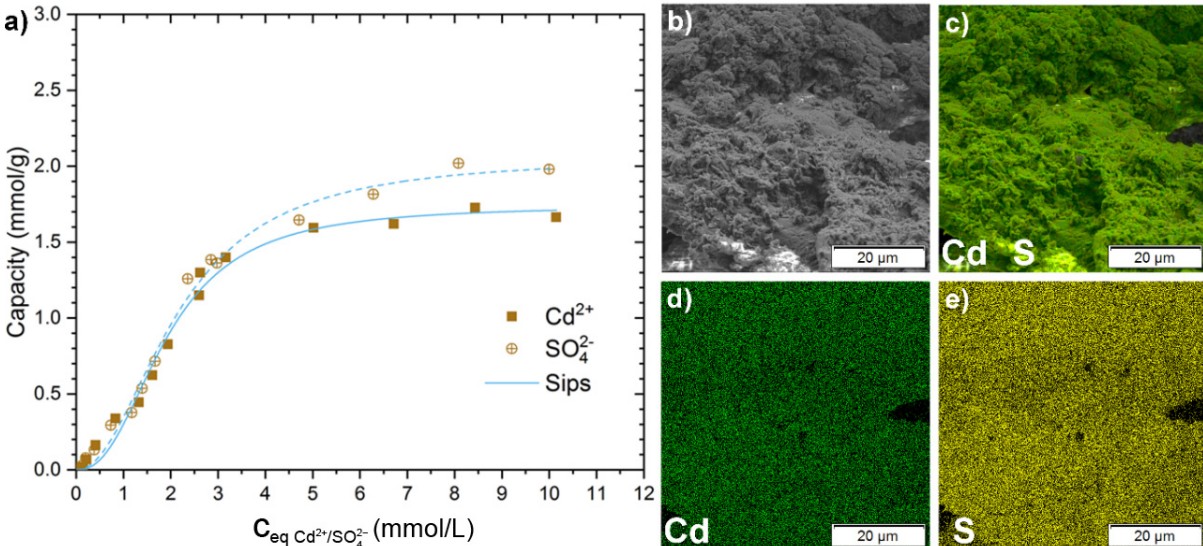

**Figure 6.** (**a**) Adsorption capacity for $Cd^{2+}$ (filled squares) and $SO_4^{2-}$ (crossed circles) from a $CdSO_4$ solution at the respective equilibrium concentrations $c_{eq}$ with the corresponding Sips isotherm model fits (a blue, solid line for $Cd^{2+}$, and a dashed line for $SO_4^{2-}$). The corresponding pH values for the experiments can be seen at Figure S19. (**b–e**) SEM-EDX image and mapping of H-Ch60-2 after the adsorption experiment with $CdSO_4$ solution ($c_0$ = 800 mg/L $Cd^{2+}$). (**b**) SEM image of H-Ch60-2 after the adsorption with $CdSO_{4(aq)}$. (**c–e**) SEM-EDX elemental mapping of (**c**) Cd (green) and S (yellow), (**d**) Cd (green), and (**e**) S (yellow). The corresponding EDX spectra are shown in Figure S21. The SEM-EDX and SEM images for H-Ch60-2 before the adsorption process are shown in Figure S8.

**Table 4.** Fitting parameters for Langmuir, Sips and Dubinin–Radushkevich isotherm models for the adsorption from a $CdSO_4$ solution. Here, $Q_m$ is the maximal sorption capacity, K is the equilibrium constant for the respective model, n is the Sips isotherm model exponent, and $E_{ads,DR}$ is the mean free energy of adsorption derived from the Dubinin–Radushkevich model. $R^2$ corresponds to the coefficient of determination (COD).

| Ion | Model | $Q_m$ mmol/g | K * | $\beta_{DR}$ $10^{-9}$ mol²/J² | n | $E_{ads,DR}$ kJ/mol | $\Delta G^\circ$ kJ/mol | $R^2$ (COD) |
|---|---|---|---|---|---|---|---|---|
| | Langmuir | 2.48 ± 0.24 | 0.28 ± 0.06 | – | – | – | −5.28 | 0.961 |
| $Al^{3+}$ | Sips | 1.74 ± 0.05 | 0.21 ± 0.03 | – | 2.41 ± 0.26 | – | – | 0.991 |
| | Dubinin-Radushkevich | 5.21 ± 0.82 | – | 7.61 ± 0.88 | – | 8.11 ± 0.47 | – | 0.938 |
| | Langmuir | 3.17 ± 0.32 | 0.21 ± 0.04 | – | – | – | −5.01 | 0.971 |
| $SO_4^{2-}$ | Sips | 2.06 ± 0.07 | 0.20 ± 0.02 | – | 2.10 ± 0.19 | – | – | 0.993 |
| | Dubinin-Radushkevich | 6.86 ± 0.98 | – | 8.54 ± 0.81 | – | 7.65 ± 0.36 | – | 0.957 |

* The unit of K for the Langmuir model is L/mmol, and for the Sips model it is $(L/mmol)^n$.

From the $CdSO_4$ adsorption experiment, the SEM-EDX analysis can be seen in Figure 6. A homogenous distribution over the whole surface of H-Ch60-2 of the elements Cd and S without clustering was found. Therefore, a homogenous and simultaneous adsorption of $Cd^{2+}$ and $SO_4^{2-}$ was concluded. The derived elemental ratios from EDX spectra (see Table 5) show a higher amount of $SO_4^{2-}$ than the respective metal ions when compared to the data from the adsorption capacities. This can be due for an increased deposition of sulfate on the outer surface while the metal ions are potentially adsorbed also into deeper layers of the chitosan.

**Table 5.** Elemental distribution in the atom% measured by SEM-EDX from pure H-Ch60-2, and H-Ch60-2 after adsorption experiments with $Al_2(SO_4)_3$ ($c_0$ = 370 mg/L $Al^{3+}$) and $CdSO_4$ ($c_0$ = 800 mg/L $Cd^{2+}$) solutions, respectively. The corresponding EDX spectra can be found in Figures S18 and S21.

| Sample | C in Atom% | N in Atom% | O in Atom% | S in Atom% | Al in Atom% | Cd in Atom% | Total in Atom% |
|---|---|---|---|---|---|---|---|
| H-Ch60-2 | 49.5 | 15.4 | 35.1 | 0 | 0 | 0 | 100 |
| H-Ch60-2 Al2(SO$_4$)$_3$ | 39.5 | 11.6 | 44.9 | 3.1 | 0.9 | 0.0 | 100 * |
| H-Ch60-2 CdSO$_4$ | 44.3 | 7.6 | 40.0 | 3.5 | 0 | 4.6 | 100 |

* F detected and excluded as a minor impurity (see Figure S18).

When comparing the achieved adsorption capacities to different biobased-adsorbents from the literature [7,11,41–52], H-chitosan shows extremely high adsorption capacities for $Cd^{2+}$ and $Al^{3+}$ (see Tables S3 and S4). In direct comparison to unmodified chitosan, an increase of the adsorption capacity by around 20% for $Cd^{2+}$ [50] and by around 400% for $Al^{3+}$ shows the significant effect of the substitution onto the adsorption process even at low substitution degrees [42].

From the adsorption isotherms, it can be seen that the affinity towards sulfate uptake of the adsorbent is higher than the respective metal ion. This is due to the facile protonation of the adsorbents' amino groups, resulting in highly positive charge and therefore good ionic interaction with the sulfate, which displaces the formed hydroxyl ions, ensuring the overall charge neutrality of the solution. Nevertheless, the sulfate uptake is significantly influenced by the present metal ion, as seen by the difference in the uptake values.

## 4. Conclusions

In summary, we could prove a drastic improvement of the maximum adsorption capacity of chitosan by partially substituting chitosan with hexanoyl chloride at the example of $CdSO_{4(aq)}$ and $Al_2(SO_4)_{3(aq)}$. The isotherms from batch sorption experiments fitted with the Sips model rendered adsorption capacities of 1.7 mmol/g for $Cd^{2+}$ with for 2.1 mmol/g $SO_4^{2-}$ and 1.8 mmol/g for $Al^{3+}$ with 2.6 mmol/g for $SO_4^{2-}$. We found that even a small amount of hexanoyl chloride substitution is fully sufficient to improve the adsorption properties of pure chitosan for water treatment applications, thereby reducing fabrication costs, even while increasing the adsorption efficiency up to 400% in the case of $Al^{3+}$. Simultaneously, the modification of the chitosan biopolymer resulted in a decrease in water solubility at low pH values, making it a promising environmentally friendly and biopolymer-based absorber for highly acidic water, which until now could only insufficiently covered by bio-based absorbers. The results shown are a further step towards a potential application of modified chitosan for the efficient and eco-friendly remediation of highly contaminated surface waters from acid mine drainage.

**Supplementary Materials:** The following supporting information can be downloaded at: https://www.mdpi.com/article/10.3390/polysaccharides3030035/s1. Scheme S1: Substitution route of chitosan Ch90-60-A1 and Ch90-200-A1 in order to produce substituted chitosan samples with hexanoyl chloride of different substitution degrees. Figure S1: $^1$H NMR spectra of the chitosan Ch90-60-A1 (purple) and its modifications H-Ch60-2 (blue), H-Ch60-8 (orange) and H-Ch60-12 (black). Figure S2: $^1$H NMR spectra of the chitosan Ch90-200-A1 (light purple) and its modifications: H-Ch200-2 (light blue), H-Ch200-8 (yellow) and H-Ch200-12 (gray). Figure S3: FTIR spectra of Ch90/60/A1 (purple), H-Ch60-2 (dark blue), H-Ch60-8 (orange). Figure S4: FTIR spectra of Ch90/200/A1 (light purple), H-Ch200-2 (light blue), H-Ch200-8 (yellow). Figure S5: TGA analysis of (a) H-Ch60-x compared with native Ch90/60/A1 and (b) H-Ch200-x compared with native Ch90/200/A. Figure S6: Nitrogen sorption measurements of Ch90/60/A1 and its derivates. All SSA are situated between 2 and 4 m$^2$/g (calculated using BET-model). Figure S7: Nitrogen sorption measurements of Ch90/200/A1 and

its derivates All SSA are situated between 2 and 5 m$^2$/g (calculated using BET-model). Figure S8: SEM-EDX of H-Ch60-2. Figure S9: SEM-EDX of H-Ch60-2, elemental distribution. Figure S10: DLS measurements of Ch90/60/A1 and its derivates with intensity and volume distribution (left to right). Figure S11: DLS measurements of Ch90/200/A1 and its derivates with intensity and volume distribution (left to right). Figure S12: pH values before (pH$_0$, shown as light green bars) and after (pH$_{eq}$, shown as dark green bars) the sorption process of Al$_2$(SO$_4$)$_3$ solution onto Ch90/60/A1, Ch90/200/A1, and their respective modifications with different weight ratios of hexanoyl chloride with an initial concentration of 20 mg/L Al$^{3+}$. The respective percentage adsorption results can be seen at Figure 4. Figure S13: pH values before (pH$_0$, shown as light green bars) and after (pH$_{eq}$, shown as dark green bars) the sorption process of Al$_2$(SO$_4$)$_3$ solution onto Ch90/60/A1, Ch90/200/A1, and their respective modifications with different weight ratios of hexanoyl chloride with an initial concentration of 2 mg/L Al$^{3+}$. The respective percentage adsorption results can be seen at Figure 4. Figure S14: pH values before (pH$_0$, shown as light brown bars) and after (pH$_{eq}$, shown as dark brown bars) the sorption process of CdSO$_4$ solution onto Ch90/60/A1, Ch90/200/A1, and their respective modifications with different weight ratios of hexanoyl chloride with an initial concentration of 0.3 mg/L Cd$^{3+}$. The respective percentage adsorption results can be seen in Figure 3. Figure S15: pH values before (pH$_0$, shown as light brown bars) and after (pH$_{eq}$, shown as dark brown bars) the sorption process of CdSO$_4$ solution onto Ch90/60/A1, Ch90/200/A1, and their respective modifications with different weight ratios of hexanoyl chloride with an initial concentration of 0.03 mg/L Cd$^{3+}$. The respective percentage adsorption results can be seen in Figure 4. Figure S16: pH values before (pH$_0$, shown as solid squares) and after (pH$_{eq}$, shown as crossed circles) the sorption process of CdSO$_4$ solution onto H-Ch60-2 at different initial concentrations. The respective adsorption results can be seen at Figure 5. Figure S17: Adsorption capacity for Al$^{3+}$ (filled squares) and SO$_4$$^{2-}$ (crossed circles) from Al$_2$(SO$_4$)$_3$ solution at the respective equilibrium concentrations c$_{eq}$ with the corresponding Langmuir (black line), Sips (blue line) and Dubinin–Radushkevich (orange) isotherm model fits with solid lines used for fits of Al$^{3+}$, and a dashed line for SO$_4$$^{2-}$. Figure S18: EDX spectrum of H-Ch60-2 after the adsorption experiment with Al$_2$(SO$_4$)$_3$ solution (c$_0$ = 370 mg/L Al$^{3+}$). Figure S19: pH values before (pH$_0$, shown as solid squares) and after (pH$_{eq}$, shown as crossed circles) the sorption process of CdSO$_4$ solution onto H-Ch60-2 at different initial concentrations. The respective adsorption results can be seen at Figure 5. Figure S20: Adsorption capacity for Cd$^{2+}$ (filled squares) and SO$_4$$^{2-}$ (crossed circles) from CdSO$_4$ solution at the respective equilibrium concentrations c$_{eq}$ with the corresponding Langmuir (black line), Sips (blue line) and Dubinin–Radushkevich (orange) isotherm model fits, with solid lines used for fits of Cd$^{2+}$, and a dashed line for SO$_4$$^2$. Figure S21: EDX spectrum of H-Ch60-2 after the adsorption experiment with CdSO$_4$ solution (c$_0$ = 800 mg/L Cd$^{2+}$). Table S1: Isoelectric points (IEP) of the H-chitosan and pure chitosan samples obtained from the streaming potential vs. pH curves shown in Figure 2d,e. Table S2: Sorption capacities for the removal of aluminum compounds from aqueous solutions with different sorbent materials. Table S3: Sorption capacities for the removal of cadmium compounds from aqueous solutions with different sorbent materials.

**Author Contributions:** Conceptualization, B.R., K.B.L.B. and D.S.; methodology, B.R., K.B.L.B. and S.P.; validation, B.R. and K.B.L.B.; formal analysis, B.R.; investigation, B.R., K.B.L.B., M.K., M.M., K.H.C., C.S., N.G. and R.B.; resources, S.S. and D.S.; data curation, B.R., K.B.L.B. and M.K.; writing—original draft preparation, B.R. and K.B.L.B.; writing—review and editing, B.R., K.B.L.B., R.B., M.K., K.H.C., C.S., N.G., S.S., S.P. and D.S.; visualization, B.R., K.B.L.B. and D.S.; supervision, S.S. and D.S.; project administration, S.S. and D.S.; funding acquisition, S.S. and D.S. All authors have read and agreed to the published version of the manuscript.

**Funding:** The authors gratefully acknowledge the support received from the Saxonian funding organizations SAB and LfULG within the cooperative projects entitled "Development of environmentally compatible biopolymers as flocculants/adsorbents for the removal of iron and sulfate ions from surface waters" (SAB no 100377122) and "Mobile sensor systems for on-site heavy metal detection in water" (contract 33-8128/157/1), respectively. Furthermore, we acknowledge the German research foundation BMBF (Federal Ministry for Education and Research), project ISOMAT (project No. 01DS18022).

**Institutional Review Board Statement:** Not applicable.

**Informed Consent Statement:** Not applicable.

**Data Availability Statement:** The data presented in this study are available in the article or the Supplementary Materials.

**Acknowledgments:** The authors thank BioLog® Heppe GmbH from Germany for the support of the materials, and the discussions and cooperation. Furthermore, we thank Hartmut Komber for the NMR measurements and fruitful discussions. For the determination of the molecular weight of the Ch90/200/A1 via the AF4 method, we want to thank Susanne Boye. Birgit Urban is thanked for the ATR-FTIR measurements of the unmodified and hexanoylated chitosan films.

**Conflicts of Interest:** The authors declare no conflict of interest. The funders had no role in the design of the study; in the collection, analyses, or interpretation of data; in the writing of the manuscript; or in the decision to publish the results.

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
