# Peer review of "Ecofriendly Removal of Aluminum and Cadmium Sulfate Pollution by Adsorption on Hexanoyl-Modified Chitosan"

_2673-4176, doi:10.3390/polysaccharides3030035_

Round 1

Reviewer 1 Report

This is a very good paper, well-written, with a well-designed experiments and the discussion is a logical outcome of experimental data. I recommend acceptance of this ms. I only have a couple of issues that I would like authors comment on.

The sentence “chitosan was modified with 20 hexanoyl chloride (H-chitosan) to reduce the solubility.” should be clarified to avoid misunderstandings.

The sorption isotherms seem to be discussed in terms of salts and not cation plus anion independently (see line 459); however, it seems that the material is more effective for sulfate rather than metal ions (e.g., 7.80 mmol for sulfate and 3.52 mmol for Al(III) are adsorbed – taking into account the salt stoichiometry), which also compromise the electroneutrality of solution / material. This should be comment by authors.

Reviewer 2 Report

The manuscript, entitled “Ecofriendly Removal of Aluminum and Cadmium Sulfate Pollution by Adsorption on Hexanoyl Modified Chitosan” is an interesting paper. According to me, this manuscript merits to be accepted in the journal Polysaccharides. However, the below changes should be addressed before taking it further.

·        The novelty of this research should be inserted in the text clearly.

·        Adsorption experiments with heavy metal salts; more experimental details must be added

·        Adsorption kinetics must be added to give more information about the adsorption mechanism.

·        The adsorption thermodynamics may be investigated in details.

·        Realated to BET surface area, the samples have micropore volume ? please, put the BJH figure and also , make one table for differeniate the samples in terms of surface area , pore volume , micropore volume , mesopore volume?

·        Authors haven't studied the reusability of adsorbent. Why?

Reviewer 3 Report

This draft describes the chemical modification of chitosan by reaction with hexanoyl chloride. Modified chitosans are used for water purification treatment by adsorption of Cd2+ and Al3+ ions. This paper can be published after minor revision.

Page 1, Line 24: avoid to use abreviations in the abstract (H-Ch60-2)

Table 1: add a column with hexanoyl chloride added in mol

At the begining of section Result and Discussion, the authors could add a scheme with the chemical reaction.

For dynamic light scattering, size distributions in intensity are preferred instead size distributions in number.

Round 2

Reviewer 1 Report

Comente were properly addressed.

Reviewer 2 Report

The paper can be accepted in the present form.